# Intelligent Identification of Maceral Components of Coal Based on Image Segmentation and Classification

**Hongdong Wang [1], Meng Lei [1,2]**, **Yilin Chen [3], Ming Li [1] and Liang Zou [1,2],***

[1] School of Information and Control Engineering, China University of Mining and Technology, Xuzhou 221116, China

[2] Department of Electrical and Computer Engineering, University of British Columbia, Vancouver, BC V6T 1Z4, Canada

[3] School of Resources and Geosciences, China University of Mining and Technology, Xuzhou 221116, China

* Correspondence: liangzou@ece.ubc.ca

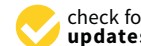

**Featured Application: Maceral Analysis; Coal Processing.**

**Abstract:** An intelligent analytical technique which is able to accurately identify maceral components is highly desired in the fields of mining and geology. However, currently available methods based on fixed-size window neglect the shape information, and thus do not work in identifying maceral composition from one entire photomicrograph. To address these concerns, we propose a novel Maceral Identification strategy based on image Segmentation and Classification (MISC). Considering the complex and heterogeneous nature of coal, a two-level coarse-to-fine clustering method based on K-means is employed to divide microscopic images into a sequence of regions with similar attributes (i.e., binder, vitrinite, liptinite and inertinite). Furthermore, comprehensive features along with random forest are utilized to automatically classify binder and seven types of maceral components, including vitrinite, fusinite, semifusinite, cutinite, sporinite, inertodetrinite and micrinite. Evaluations on 39 microscopic images show that the proposed method achieves the state-of-the-art accuracy of 90.44% and serves as the baseline for future research on maceral analysis. In addition, to support the decisions of petrologists during maceral analysis, we developed a standalone software, which is freely available at https://github.com/GuyooGu/MISC-Master.

**Keywords:** maceral components; image segmentation; coal petrography; random forest; two-level clustering

## 1. Introduction

### 1.1. Background and Motivation

Coal is an extremely complex heterogeneous material formed from ancient wetlands over geological processes. It consists of various organic components called macerals and a lesser amount of inorganic minerals [1,2]. Different from the minerals with homogeneous internal composition and structures, the macerals derived from coalified plant tissues have distinct physical and chemical properties, and are related to the degree of coalification. In addition, the maceral composition is an important factor in evaluating the coal seam quality. Precise identification of the maceral components has a multitude of uses across various industry sectors, including hydrogenation, combustion, carbonization and gasification [2–4].

Macerals can be categorized into three basic groups through petrographic analysis, including vitrinite from coalified woody tissue, liptinite from more decay-resistant parts of plants and inertinite

from hydrogen-rich plant and decomposition products. These maceral groups are subdivided into maceral subgroups and macerals [5]. Most laboratories associated with the coke-making industry Standard Test Method for Microscopical Determination of the Maceral Composition follow the standard test methods, such as International Commission for Coal Petrology (ICCP) standard [6] and American Society for Testing and Materials (ASTM) standard D2799-13 [7], and the microscopic analysis is always performed manually for the identification of maceral components. Despite being the most widely used method, it is costly and labor-intensive to identify the maceral composition due to the complicated nature and substantial diversification of the petrographical properties of coal. Even for specialists in petrography, they may arrive at different judgements in the analysis of the same microscopic image as a consequence of the subjective factors. To address these concerns, an intelligent analytical technique which is able to automatically provide objective identification of maceral components is highly desired for the growing industrial demand.

### 1.2. Related Work

Automatic geological identification is becoming an increasingly important technique in various fields, such as in mining and geology. Camalan et al. presented a novel strategy to estimate the liberation spectrum from optical micrographs via random forest [8]. Lei et al. proposed an autonomous classification method of rock images via unsupervised feature learning [9]. In [10], transfer learning was employed to deal with the problem of cross-region microscopic sandstone images classification. Numerous attempts have been made on the classification of microscopic rock images and achieved great success [11–13]. Considering the heterogeneous nature of coal, automatic identification of maceral components is still a challenging task [3].

In the early stage, the analytical methods to estimate the volume proportions of coal macerals were mainly based on the gray scale value of pixels [14,15] and provided interesting results. However, the liptinite and background resin have similar gray scale values, and therefore it is difficult to separate them. In addition, different maceral components in a maceral group differ only subtly in term of the reflectance, and the existing methods merely based on gray scale values are not suitable for distinguishing maceral components. Furthermore, the gray scale values of a specific component may vary over a large range with the degree of coalification. Although the gray scale descriptions of pixels remain important, the need for more quantitative features, such as shape and texture, from photomicrographs has been recognized. With the development of machine learning and image processing, it is possible to automatically make more elaborate classifications [12,16]. Over the past several years, attempts based on machine learning techniques have shown promising results in maceral analysis.

Wang et al. utilized principal component analysis (PCA) to extract primary features from texture-related and intensity-related features, and employed Support Vector Machine (SVM) to classify maceral components of the vitrinite group [17]. Skiba et al. selected 10 textural features via PCA and developed a novel strategy for automatic identification of macerals of the inertinite group. The proposed method achieved an outstanding accuracy of 93.6% based on a group of neural networks [16]. Most of the works focus on a single maceral group. So far, attempts to provide full identification of macerals have been considerably limited. Młynarczuk and Skiba evaluated the ability of three machine learning methods for identifying three maceral groups of coal and non-organic minerals [3] . They cropped the region of interest (ROI) of 41 px × 41 px and determined the label of the central pixel. Considering the morphological gradients along with the gray level features, the proposed method achieved an accuracy of 97% in classifying maceral groups. Furthermore, they analyzed six kinds of macerals of the inertinite group and an obtained accuracy is over 91%. Pearson and CSIRO have released two automated tools to identify maceral composition, including Pearson Coal Petrography (Pearson Coal Petrography—http://www.coalpetrography.com/blog1/) and CSIRO coal grain analysis (CGS) (Coal Grain Analysis—https://www.csiro.au/en/Do-business/Commercialisation/

Marketplace/Coal-Grain-Analysis), which have been successfully commercialized. However, they do not mention the detailed technologies employed in these two tools on the corresponding websites.

Despite providing inspiring performance via machine learning-based methods, there are many issues that require more scientific breakthroughs. The motivations of the proposed method derive from the following three aspects:

First and foremost, both patch-wise classification aiming to assign a label to a given region and pixel-wise classification aiming to provide a label for the central pixel of ROI neglect the shape and the size information. For instance, the micrinites are always small in size and the cutinites are very thin [18,19]. In cases where the selected regions contain two or more groups/components of macerals, it directly affects the performance. The results of the previous methods based on fixed-size window are always observed with poor generalized ability.

Second, due to the complex and heterogeneous nature of coal, the task for identifying macerals requires more parameters describing the shape, texture and morphology. Comprehensive features along with powerful machine learning techniques are required to detect the subtle differences between maceral groups/components.

Last but not least, there is no publicly available software for identifying maceral groups/components, especially targeted for geologists without strong expertise in the machine learning and image processing domains. In addition, the existing methods focus on predicting the label for a given region or pixel, whereas they do not work in the identification of maceral components from the entire photomicrograph.

To address the above-mentioned concerns, we propose a novel framework for autonomous coal macerals identification based on image segmentation and classification (MISC). A two-level coarse-to-fine clustering strategy is implemented for image segmentation, and random forest is employed to classify maceral components from the entire microscopic image. The main contributions of the proposed framework can primarily be broken down into three aspects.

1. Inspired by the distribution of maceral subgroups and the gray scale characteristics of maceral groups, we design a coarse-to-fine segmentation strategy to divide an entire photomicrograph into a number of discrete regions, providing the shape and the size information. Both coarse clustering and fine clustering are based on K-means, which is one of the most popular unsupervised image segmentation methods.
2. We extract the discriminative features from microscopic images, including geometric, grayscale and texture features, which are combined into a 172-element feature vector. A comprehensive feature combination for identification of maceral components, not limited to maceral groups, is proposed. In addition, we evaluate six kinds of machine learning classifiers, and the random forest provides the best performance with an average accuracy of 90.44%.
3. A publicly available tool, namely MISC, to identify macerals in microscopic images of coal is released. The software integrates the best segmentation and classification algorithms involved in this paper, and provides an AI-assisted autonomy algorithm for maceral components identification.

## 2. Experiment Dataset

The metallurgical coal samples used in the study are randomly selected from samples submitted to the laboratory of the United States Geological Survey (USGS). The selected samples were prepared through a sequence of operations, including sieving, molding and polishing. All the procedures follow protocols established by the ASTM Standard D2797 [20]. Photomicrographs are captured using a Leica DMRX microscope with a Leica DFC 480 digital camera under incident white light in oil immersion, in accordance with ASTM standard D2799-13 [7].

With the increment of coalification, the difference in term of gray levels between macerals is reduced. It will be difficult to distinguish between vitrinite and liptinite at a high degree of coalification (e.g., R0 > 1.25%). In this work, the selected coal samples are with a relatively low degree of coalification (i.e., R0 < 1%). The maceral composition of each coal sample was annotated by 5 petrographers according to ASTM Standard D2799-13, and the 39 samples out of 50 samples with consistent results were further analyzed. The resolutions of these photomicrographs are different with each other, in the range of (267–1024) × (230–768) px, with each pixel roughly corresponding to 2–4 μm. Table 1 shows all the maceral components analyzed in this study, with brief descriptions and the number of macerals (909 in total). In addition, 64 objects belonging to binder are also included in the dataset. Binder is large relative to maceral components and can hold these components together. The demonstration of each maceral is provided in Figure 1.

**Table 1.** Brief description of seven types of macerals used in this study.

| Maceral Group | Maceral | Brief Description of Specific Maceral | The Number of Macerals |
|---|---|---|---|
| vitrinite | ____ | The predominate maceral in most coals of intermediate reflectance. It is always brilliantly glossy resembling vitreous. Vitrinite is derived from coalified woody tissue and occurs generally in thin bands of 2–10 mm thickness. It plays an important role in defining the properties of the whole coal. | 116 |
| liptinite | sporinite | A liptinite maceral exhibition various lenticular, oval, round forms, or small rod-like projections. | 102 |
| | cutinite | A maceral is derived from the stratum corneum of plant leaves, roots and stems. It occurs as stringers strips of varying thickness, with a smooth outer margin and serrated edges. It is not very abundant. | 30 |
| inertinite | fusinite | An inertinite maceral distinguished principally by the preservation of some features of the plant cell wall structure. It has charcoal-like structure and is commonly broken into small shards and fragments. | 198 |
| | semifusinite | It looks like fusinite in morphology. It has the largest range of reflectance. The partial size is always great than 50 μm except when occurring as a fragment within binder. | 122 |
| | inertodetrinite | Small, discrete inertinite fragments (>2 μm in size) of varying shape. Reflectance values of inertodetrinite are greater than surrounding vitrinite macerals. | 141 |
| | micrinite | Generally, micrinite is non-angular, and occurs as particles around 1 to 5 μm diameter. | 200 |

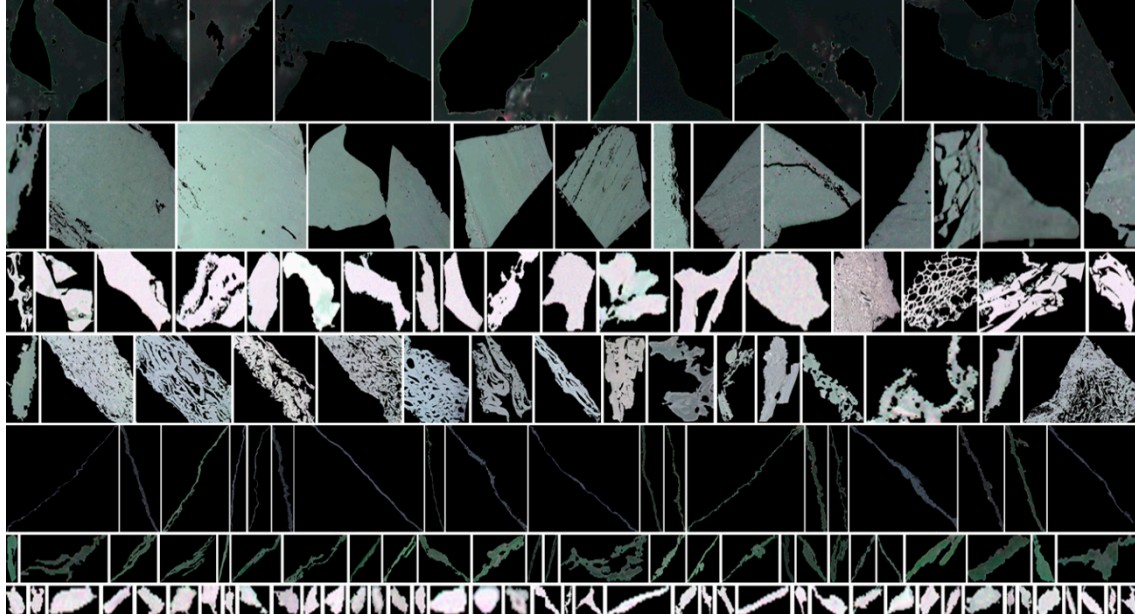

**Figure 1.** Examples of maceral components and binder. Each row represents a class. They are binder, vitrinite, fusinite, semifusinite, cutinite, sporinite and inertodetrinite from top to bottom respectively. The black color areas (i.e., RGB value = 0) in each figure represent the background of the given maceral component. The micrinite is not shown here for its small size accounting for only a few pixels.

## 3. Methods

### 3.1. Image Segmentation Based on Two-Level Clustering

The main differences between maceral groups are the gray scale values. Generally, the gray scale values of liptinite, vitrinite and inertinite decrease successively, as demonstrated in Figure 2a. The gray scale distribution curves illustrate that there are noticeable differences among three maceral groups and the binder. However, it is difficult to define the boundaries between them. In addition, the boundaries corresponding to different photographs are different. The gray scale range of each maceral group varies with the degree of coalification. Figure 2b shows the distribution of gray scale values of binder and inertinite across four photomicrographs. It can be seen that the gray range of binder is relatively fixed, whereas that of inertinite group of 4 coal samples differs greatly. Therefore, it is unreliable to adopt a fixed threshold to segment microscopic photographs with different coalification degrees.

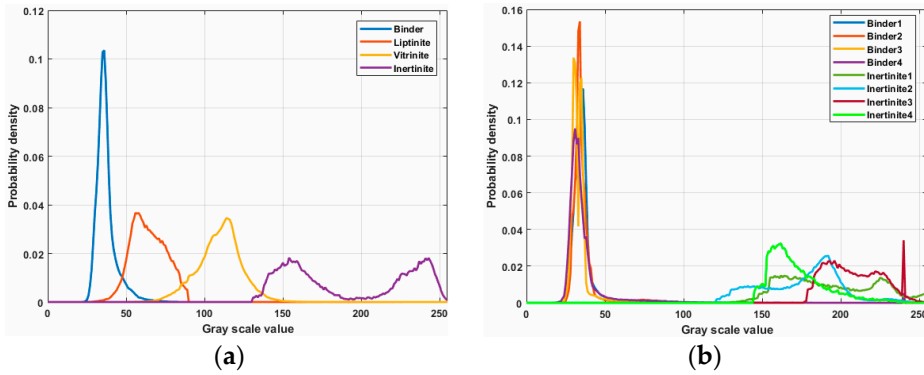

**Figure 2.** Comparison of gray scale distributions corresponding to different maceral groups and the binder. (**a**) Gray scale distributions of maceral groups and the binder in one coal sample; (**b**) the difference in the gray scale distributions across 4 coal samples. For simplicity, we only show the curves corresponding to binder and inertinite in (**b**).

In order to automatically detect the boundaries of each maceral, we employed image-wise segmentation which is able to divide each microscopic image into a sequence of discrete regions, each having similar attributes. It is an essential step for further maceral composition analysis. Considering the fact that the maceral components within each group mostly are not adjacent to each other and the gray scale values are the major difference between maceral groups, we adopt the gray scale values of each pixel as the features. K-means clustering, one of the most favorable clustering techniques, is utilized for its simplicity and computational efficiency [21,22].

The main steps of K-means algorithm can be summarized as follows:

(1) Choose initial cluster centroids $\mu_1, \mu_2, \mu_3, \ldots, \mu_k \in R^d$ randomly, where $k$ represents the number of clusters and $d$ represents the dimension of the feature space.

(2) Repeat until convergence:

For each pixel $p$, assign it to its nearest centroid,

$$c^p := \underset{j}{\operatorname{argmin}} \|\mathbf{v}^p - \mu_j\|^2 \tag{1}$$

Update each centroid,

$$\mu_j := \frac{\sum\limits_{i=1}^{M} 1\{c^p = j\}\mathbf{v}^p}{\sum\limits_{i=1}^{M} 1\{c^p = j\}} \tag{2}$$

where $c^p$ represents the cluster of pixel $p$, $\mathbf{v}^p$ is a vector consisting of RGB value of pixel $p$, and $M$ represents the total number of pixels in each photomicrograph, respectively.

In addition, due to the complexity of coal properties, it is difficult to achieve accurate clustering results by using single-level clustering only. Single-level clustering is difficult to provide satisfactory segmentation results. Hence, a coarse-to-fine strategy is adopted. In the coarse clustering level, the regular K-means clustering algorithm is first applied to get rough clustering results, which splits a whole image into two sub-clusters. In the fine clustering level, each of the previous clusters is partitioned again into two fine sub-clusters (i.e., maceral groups and binder).

### 3.2. Feature Extraction

Inspired by the way that the petrologists examine photomicrographs, we extracted three types of features for maceral identification. Table 2 lists 172 features utilized in this study, such as reflectance contrasts, shape, morphology and size, which can be categorized into geometric, grayscale and texture features [23]. The detailed descriptions of these features can refer to the papers on image pattern recognition [24,25].

**Table 2.** Feature space utilized in this study.

| Geometric Features (x1–x16) | Grayscale Features (x17–x90) | Texture Features (x91–x172) |
|---|---|---|
| x1: Area | x17: Mean gray value | x91–x92: Mean and standard deviation of energy |
| x2: Perimeter | x18: Standard deviation of gray value | x93–x94: Mean and standard deviation of entropy |
| x3: Rectangle degree | x19: Max gray value | x95–x96: Mean and standard deviation of inertial moment |
| x4: Aspect ratio | x20: Min gray value | x97–x98: Mean and standard deviation of correlative |
| x5: Length of long axis | x21: Gray scale median | X99–x100: Small and large gradient advantage |
| x6: Length of short axis | x22: Gray scale mode | x101–x102: Inhomogeneity of grayscale and gradient distribution |
| x7: Eccentricity | x23: Average contrast | x103: energy |
| x8: Solidity | x24: Smoothing Degree | x104–x105: Mean value of grayscale and gradient |
| x9: Extent | x25: Third-order Moment | x106–x107: Mean variance of grayscale and gradient |
| x10–x16: Hu' seven invariant moments | x26–x90: Grayscale probability | x108: correlative |
| | | x109–x111: Grayscale entropy, gradient entropy, mixed entropy |
| | | x112: inertia |
| | | x113: Deficit moment |
| | | x114–x172: Fifty-nine local binary pattern features |

For instance, we employ the image moment as the shape descriptor. The moment invariants have been extensively exploited to characterize image patterns. Among various image moments, Hu's 7 invariant moments have been widely applied in a variety of applications for its invariant features on image translation, scaling and rotation [26]. They are defined as follows:

$$
\begin{aligned}
M_1 &= \eta_{20} + \eta_{02}\\
M_2 &= (\eta_{20} - \eta_{02})^2 + 4\eta_{11}\\
M_3 &= (\eta_{30} - 3\eta_{12})^2 + (3\eta_{21} - \eta_{03})^2\\
M_4 &= (\eta_{30} + \eta_{12})^2 + (\eta_{21} + \eta_{03})^2\\
M_5 &= (\eta_{03} - 3\eta_{12})(\eta_{30} + \eta_{12})[(\eta_{30} + 3\eta_{12})^2 - 3(\eta_{21} + \eta_{03})^2]+\\
&\quad (3\eta_{21} - \eta_{03})(\eta_{21} + \eta_{03})[3(\eta_{30} + \eta_{12})^2 - (\eta_{21} + \eta_{03})^2]\\
M_6 &= (\eta_{20} - \eta_{02})[(\eta_{30} + \eta_{12})^2 - (\eta_{21} + \eta_{03})^2] + 4\eta_{11}(\eta_{30} + \eta_{12})(\eta_{21} + \eta_{03})\\
M_7 &= (3\eta_{21} - \eta_{03})(\eta_{30} + \eta_{12})[(\eta_{30} + \eta_{12})^2 - 3(\eta_{21} + \eta_{03})^2]+\\
&\quad (3\eta_{21} - \eta_{30})(\eta_{21} + \eta_{03})[3(\eta_{30} + \eta_{12})^2 - (\eta_{21} + \eta_{03})^2]
\end{aligned}
\tag{3}
$$

$$
\eta_{ab} = \frac{\mu_{ab}}{\mu_{00}^{\rho}}, \rho = \frac{a+b}{2} + 1
\tag{4}
$$

where $\mu_{ab}$ represents the central moment and $\eta_{ab}$ stands for the normalized central moments.

The features x17–x90 are the statistical characters related to the gray scale values of the region of interest; x91–x99 are statistics for examining texture features based on the spatial relationship of pixels [27]; x99–x113 are the gray gradient features [28]; and the remaining features x114–x172 correspond to local binary patterns encoding the texture information [29].

### 3.3. Random Forest for Image Classification

Random forest (RF) is an ensemble machine learning method, which consists of multiple uncorrelated decision trees. It is widely used in image classification tasks due to its high accuracy, easy parameterization and robustness against overfitting [30]. Figure 3 illustrates how the random forest model works. Given the dataset with $N$ samples $D = \left\{ (x^1, y^1), \ldots, (x^l, y^l), \ldots, (x^N, y^N) \right\}$, where $x^l = [x^l(1), x^l(2), \ldots, x^l(172)]$ and $y^l \in \{1, 2, 3, 4, 5, 6, 7, 8\}$ denote the input 172 features and the output label of sample $l$, the general idea of random forest can be described as,

(1) Randomly select $N$ samples with replacement from the original dataset, and obtain $N$ subsamples for constructing each tree.

(2) Select features for constructing decision tree nodes from a random subset of all 172 features, and construct a decision tree.

(3) Repeat step (1) and (2) for $B$ times and construct a random forest with $B$ trees. The final prediction result is obtained by the majority vote of the trees in the forest.

As an ensemble model, random forest model fits the input data in a shorter time as each decision tree is independent, making parallel computing and modeling possible [31,32]. We also test the performance of the other five machine learning methods, including Fine Tree, Radial Basis Function kernel Support Vector Machine (RBF SVM), Weighted K-Nearest Neighbors (KNN), Linear Discriminant and Subspace KNN [33,34].

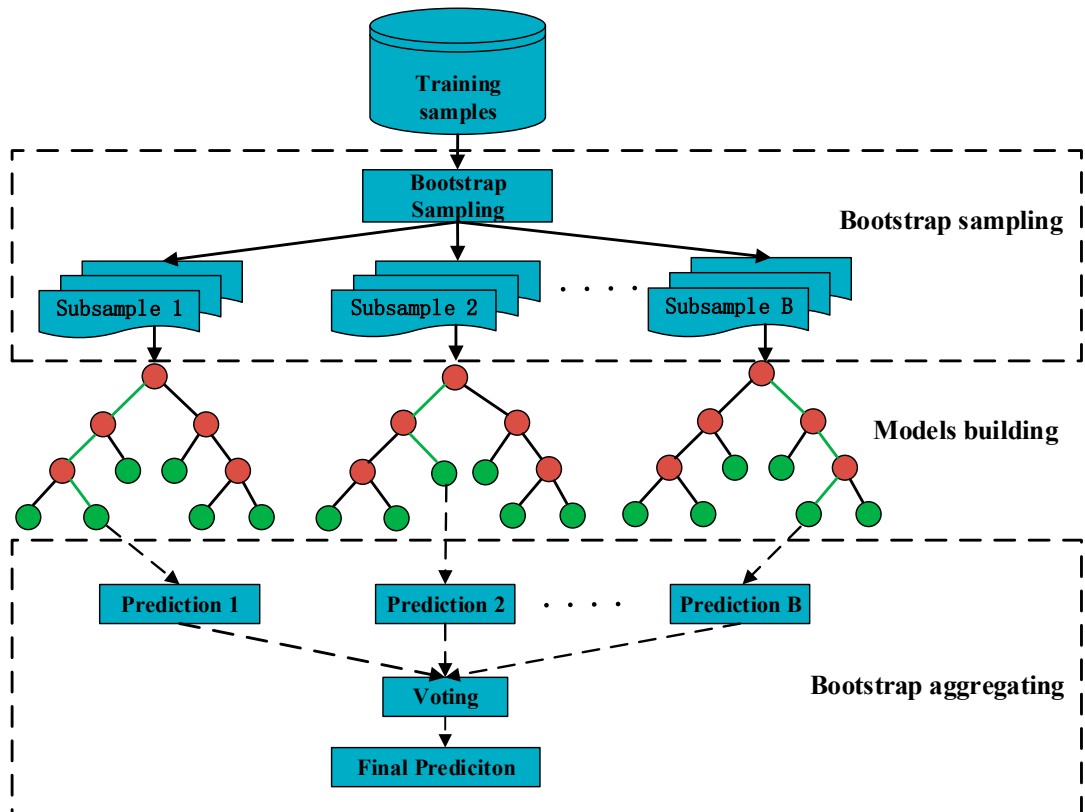

**Figure 3.** The scheme of random forest algorithm. The final prediction is obtained by taking a majority vote of the predictions from all the trees in the forest.

### 3.4. Evalutation Criteria

The results of the automated segmentation method are quantitatively evaluated by using three popular evaluation criteria including clustering accuracy, entropy and purity. We refer to the class labels as the ground truth and the results of clustering algorithms as the clusters [35,36]. We refer to class as the ground truth and cluster as the results of clustering algorithms. Clustering accuracy is the most intuitive measure to evaluate the performance of clustering, which is defined as follows

$$accuracy = \sum_{i=1}^{4} \frac{n_{ii}}{n} \tag{5}$$

where $n_{ii}$ represents the number of common samples in cluster $i$ and class $i$, $i \in \{1, 2, \ldots, 4\}$, and $n$ is the size of the data set.

Entropy is an information theoretic measure and is defined as

$$entropy = -\sum_{i=1}^{4} \frac{n_i}{n} \sum_{j=1}^{4} \frac{n_{ij}}{n_i} \log_2 \frac{n_{ij}}{n_i} \tag{6}$$

where $n_{ij}$ indicates the number of common samples in cluster $i$ and class $j$, $n_i$ stands for the number of samples in cluster $i$.

Purity is computed to measure the degree of clusters containing a single class. The purity is calculated as follows

$$purity = \sum_{i=1}^{4} \frac{n_i}{n} \max(\frac{n_{ij}}{n_i}) \tag{7}$$

## 4. Experimental Results and Discussion

### 4.1. Image Segmenation

The proposed segmentation strategy has been tested on 39 microscopic images taken by Leica DFC 480 digital camera, and the results in terms of accuracy, purity and entropy are given in Table 3. It can be observed that the proposed two-level K-means algorithm achieved significantly higher accuracy (90.82%), higher purity (90.82%) and lower entropy (0.6042) than the other clustering algorithms. In particular, the output result via two-level coarse-to-fine clustering consistently has better segmentation results as compared to the corresponding single-level clustering. For instance, regarding the K-means algorithm, the accuracy of the two-step strategy is 17.59% better than that of applying single-level K-means.

**Table 3.** Quantitative assessment of automatic segmentation methods.

| Methods | Accuracy (%) | Purity (%) | Entropy |
|---|---|---|---|
| Fuzzy c-means | 69.35 | 86.36 | 0.7291 |
| K-medoids | 68.34 | 89.94 | 0.6805 |
| K-means | 73.21 | 89.43 | 0.6748 |
| 2-level Fuzzy c-means | 76.58 | 83.33 | 0.7961 |
| 2-level K-medoids | 82.14 | 85.31 | 0.7306 |
| 2-level K-means | 90.82 | 90.82 | 0.6042 |

Furthermore, in this paper, one out of those tested images was selected to visualize the performance of the proposed strategy and the other five kinds of clustering methods. The segmentation results of single-level clustering and two-level clustering are compared with the ground truth segmentations provided by five petrologists for evaluation. It can be seen from Figure 4 that the boundary of the resultant segmentation images by K-means is slightly clearer as compared to Fuzzy c-means (FCM) and K-medoids clustering. The segmented images produced by the two-level clustering are sharper and much closer to the ground truth. Overall, the proposed two-level coarse-to-fine clustering strategy based on K-means has outperformed the other clustering algorithms.

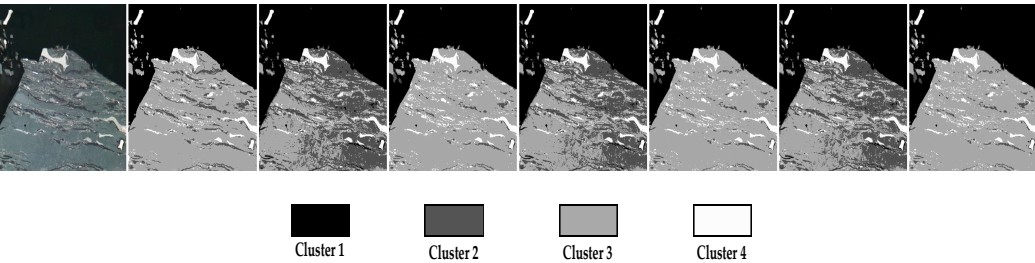

**Figure 4.** Comparison of automated segmentation results with ground truth. From left to right, the images represent original image, ground truth, the segmentation results of single-level Fuzzy c-means (FCM), 2-level FCM, single-level K-medoids, 2-level K-medoids, single-level K-means, and 2-level K-means.

We also compare the results of image segmentation with the results of the fixed-size window strategy. As can be seen from Figure 5, it is feasible to detect the thin sporinites (i.e., a-1) and the granular micrinites (i.e., a-2). We can obtain the shape information for each object of interest. As to the fixed-size window strategy, 41 px × 41 px was demonstrated to be the optimal size for maceral identification [3]. However, it is unrealistic to retrieve the shape information in feature extraction. In addition, micrinites distributing through the window may only account for a small amount of the area, and therefore it might be unreliable to train the classification models based on the information provided by the whole window. Similarly, sporinites are always thin and they might be misclassified based on the

fixed-size window strategy. Our method is also very effective in extracting maceral composition with a large area (i.e., a-3), which is helpful to improve classification accuracy. Although the proposed strategy achieved satisfying performance in identifying the objects of interest (i.e., maceral groups), the differences in term of gray scale values between maceral components are too subtle to differentiate them. More discriminative features are required.

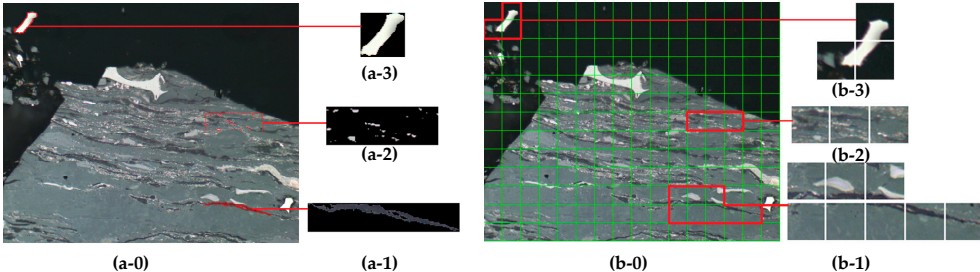

**Figure 5.** The results of image segmentation based on (**a**) the proposed Maceral Identification strategy based on image Segmentation and Classification (MISC) and (**b**) the fixed-size window strategy. (a-0) and (b-0) denote the original image. (a-1) and (b-1) are the enlargement of sporinites, (a-2) and (b-2) are the enlargement of micrinites, (a-3) and (b-3) are the enlargement of fusinites.

### 4.2. Maceral Composition Classification

Each microscopic image was divided into a series of discrete objects (i.e., macerals groups). Then we extracted geometric, grayscale, texture features from each object, and created a 172-dimensional feature vector for each object. We compared the recognition performance of random forest with five popular classification methods via a 10-fold cross validation. The experiments were repeated 10 times and the average accuracies were reported in Table 4. This summarizes the classification results of all the 973 objects obtained by image segmentation. The proposed approach yields a high accuracy of 90.44%, outperforming other classifiers. This should be regarded as a very good performance, especially when we take into account the high degree of complexity of coals. The obtained results show that the proposed strategy based on image segmentation and classification has a high potential for maceral components identification.

**Table 4.** Comparing the results of random forest and the other 5 machine learning methods for identifying maceral components (10-fold cross-validation, and repeat the experiment 10 times).

| | Classification Accuracy (%) ± std (%) | | | |
|---|---|---|---|---|
| **Classifier** | **Geometric Features** | **Grayscale Features** | **Texture Features** | **All Features** |
| Fine Tree | 72.29 ± 0.72 | 70.01 ± 0.83 | 78.79 ± 1.02 | 85.71 ± 1.08 |
| Radial basis function kernel support-vector machine | 72.47 ± 0.11 | 76.97 ± 0.59 | 85.54 ± 0.33 | 86.62 ± 0.58 |
| Weighted K-Nearest Neighbors | 55.22 ± 0.49 | 71.40 ± 0.56 | 80.44 ± 0.47 | 78.69 ± 0.88 |
| Linear Discriminant | 53.76 ± 0.32 | 70.34 ± 0.46 | 82.12 ± 0.40 | 83.36 ± 0.60 |
| Subspace K-Nearest Neighbors | 55.45 ± 0.74 | 71.13 ± 0.68 | 79.61 ± 0.56 | 81.26 ± 0.47 |
| Random Forest | 79.30 ± 0.47 | 78.86 ± 0.34 | 85.64 ± 0.30 | 90.44 ± 0.37 |

We further test the performance of an individual kind of features. The recognition performance using texture features is much higher than that of the other two types of features. Generally, the fusion of multiple kinds of features can achieve a significant improvement over a single kind of features, except for weighted KNN classifier.

To observe relations between the predictions of classifiers and the true labels, we also employ confusion matrices to report the results of different approaches. The confusion matrix enables us to know not only the error rates being made by a classifier but also the types of errors. More specifically, the rows of the matrix represent the predicted class, and the columns correspond to the true class (i.e., ground truth). The green diagonal cells stand for the number of correctly classified observations, while the red

cells represent the misclassified observations. The precision and the recall rate corresponding to each class are also shown at the far right of each row and the bottom of each column, respectively. The overall accuracy is shown at the bottom-right corner of the matrix. As shown in Figure 6, the proposed method provides satisfying performance for most of the maceral components. The main flaws of all these six classifiers come from the misclassification of semifusinite and inertodetrinite. The following reasons could contribute to the worse performance for these two components: semifusinite and inertodetrinite belong to inertinite group, and the difference in terms of gray level is too subtle to classify them properly [37]; semifusinite is intermediate between fusinite and vitrinite, and has a similar texture and general structure to fusinite; the origin of inertodetrinite is similar to fusinite and semifusinite [38].

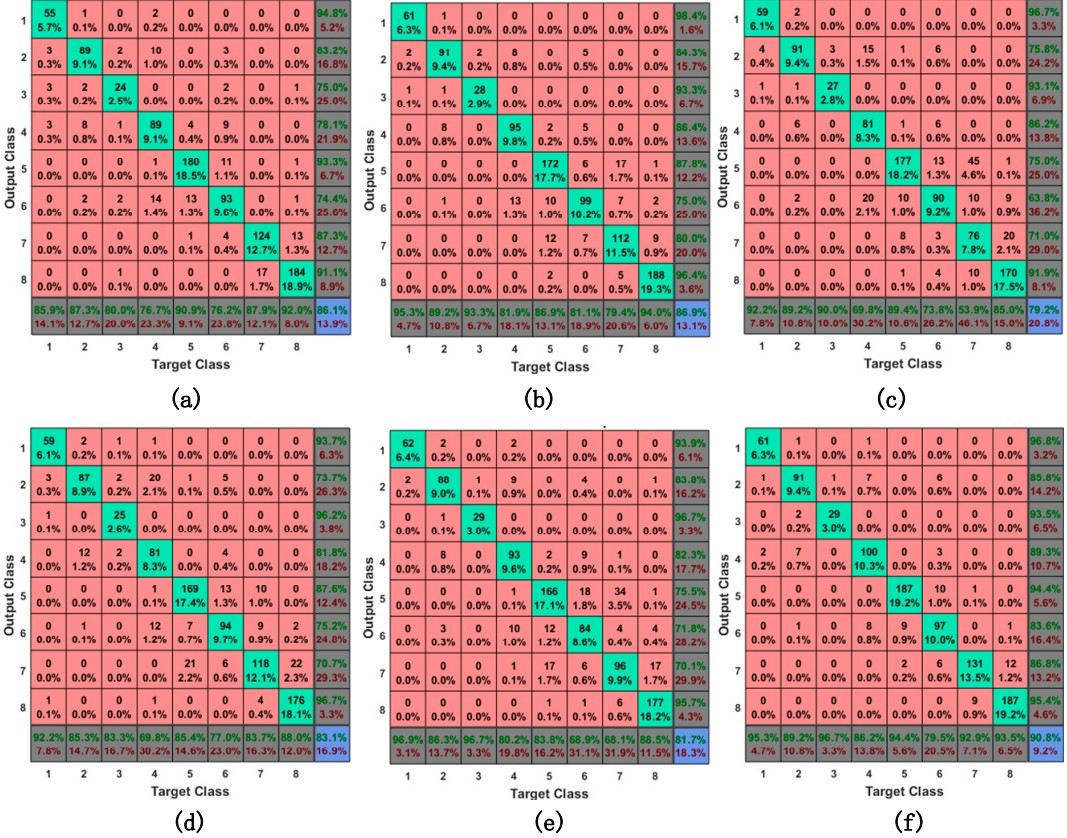

**Figure 6.** Confusion matrix comparison of six classifiers, including the results of (**a**) Fine Tree; (**b**) Radial basis function kernel support vector machine; (**c**) Weighted K-Nearest Neighbors (KNN); (**d**) Linear Discriminant Analysis; (**e**) Subspace KNN and (**f**) Random Forest. The labels in each subfigure include 1: binder, 2: sporinite, 3: cutinite, 4: vitrinite, 5: fusinite, 6: semifusinite, 7: inertodetrinite, and 8: micrinite.

The process of training a random forest involves the construction of multiple decision trees. In this study, we also evaluated the classification accuracy with the increase of the number of trees from 1 to 300. As shown in Figure 7, in general, more trees can provide the better performance, especially when the number of trees is smaller than 50. However, the improvement decreases as the number of trees increases from 50. Considering the tradeoff between the computational efficiency and the robustness of the developed model, in this study, we set the number of trees to be 200.

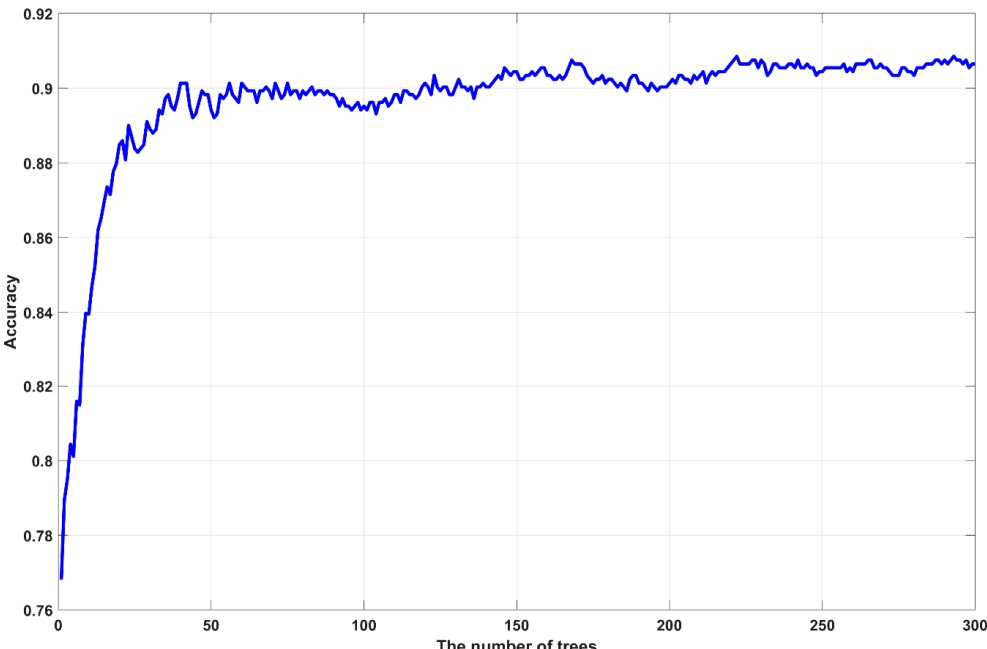

**Figure 7.** The identification accuracies with different number of trees in random forest.

### 4.3. The Platform of Automatic Coal Petrographic Analysis

The proposed maceral analysis method, MISC, based on image segmentation and classification makes it possible to identify the maceral composition automatically and intelligently. In order to facilitate the usage by petrologists, we developed a standalone software implemented in Matlab. We integrate the two-level K-means and various classification algorithms into the software for intelligent identification of maceral composition. Figure 8 is the screen snapshot of MISC software. Users can submit a microscopic image of coal with the degree of coalification R0 < 1.0%. The segmentation results are presented as four subfigures, corresponding to the binders and three maceral groups. The classification result for each object detected by image segmentation is shown with different colors for visualization. The MISC is freely available for users at the following website: https://github.com/GuyooGu/MISC-Master. It can be used to support the decisions of petrologists in classifying maceral components. To the best of our knowledge, it is the first non-commercial software for identification of maceral components. It is an efficient and effective tool for the complete analysis of maceral composition.

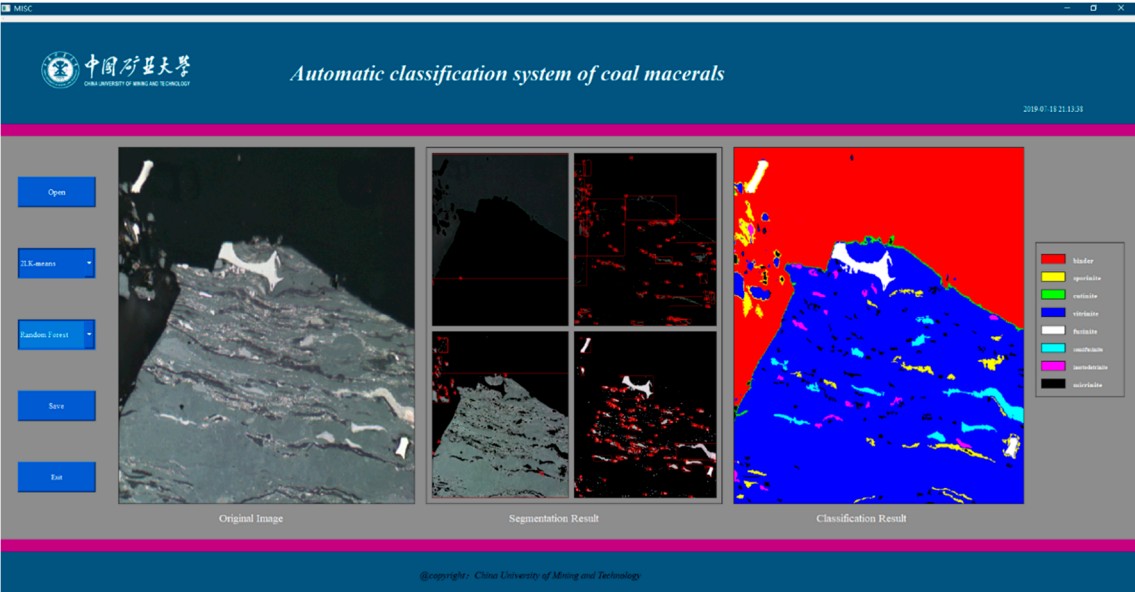

**Figure 8.** The user interface of MISC for automatic coal petrographic analysis.

## 5. Conclusions

Inspired by the way that petrologists examine photomicrographs, we proposed an automatic and effective framework for maceral classification. The proposed strategy is fundamentally different from previous attempts to classify a region and classify the central pixel of a ROI. Based on the image segmentation, rather than the fixed-size window, the regions of interest are cropped automatically. Current fixed-size window-based strategies, including both patch-wise classification and pixel-wise classification, neglect the shape and the size information and therefore do not work in identifying maceral composition from one entire photomicrograph. The utilized two-level coarse-to-fine clustering strategy achieved significantly better segmentation results as compared to the corresponding single-level clustering. In addition, considering the complicated nature of coal, it may prove difficult to identify maceral components based on a single kind of features. Our results suggest that classification approach based on multiple kinds of features, such as geometric, grayscale and texture features, will be a promising direction for identifying maceral components.

However, it should be stressed that the research described in this paper is a preliminary study which still has some limitations. First, the proposed method only works when the degree of coalification is smaller than 1.0%. With the increase of that degree, it will be difficult to distinguish vitrinite and liptinite based on gray scale values. Second, we assume that there are four maceral groups of each sample. However, this assumption may not hold in a few cases. We have tried density-based clustering methods, such as density-based spatial clustering of applications with noise (DBSCAN), which can automatically detect the optimal number of clusters. However, the performance is not better than the proposed method. We will investigate this issue in the future. In addition, 39 photomicrographs used in this study were analyzed according to definitions in ASTM Standard D 2799-13 and atlas in [39]. There are seven types of maceral components, belonging to three maceral groups (i.e., vitrinite, liptinite and inertinite). Liptinites include sporinite and cutinite. Inertinite macerals include fusinite, semifusinite, inertodetrinite, and macrinite. However, the mineral components and the other macerals, such as funginite or macrinite, are rarely found in these photomicrographs, and are not abundant in nature. Therefore, in this study, we do not consider these components, and follow a simplified classification as that in [39]. The proposed MISC strategy obtained relatively good performance, especially when we take into account a high degree of complexity of coals. Although with some limitations, to the best of our knowledge, our work is the first study aiming to provide complete analysis of maceral composition.

**Author Contributions:** Conceptualization, H.W. and L.Z.; funding acquisition, M.L. (Meng Lei) and L.Z.; methodology, H.W. and Y.C.; project administration, M.L. (Meng Lei); resources, M.L. (Ming Li); supervision, M.L. (Ming Li) and L.Z.; validation, Y.C. and M.L. (Ming Li); writing—original draft, H.W.; writing—review and editing, L.Z.

**Funding:** This research was funded by the Fundamental Research Funds for the Central Universities with grant number 2019ZDPY17.

**Conflicts of Interest:** The authors declare no conflict of interest.

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
