# Peer review of "Intelligent Identification of Maceral Components of Coal Based on Image Segmentation and Classification"

_applsci, doi:10.3390/app9163245_

Round 1

Reviewer 1 Report

The paper presents results of an interdisciplinary research work employing machine learning method to automatically identify coal maceral components from photomicrographs. The authors develop and publish a standalone software available to download. Despite being a preliminary study (works only for Ro<1.0%, this work has the potential of being a useful public imaging tool to identify the maceral concentration of coal sample. With the spirit of improving the current manuscript, I would like to suggest the authors to include the work conducted by Pearson petrography and CSIRO coal grain analysis in the author’s related work section. There two automated microscopic techniques have been successfully commercialised and do not have a limitation in coalification degree.

Furthermore, I would like to ask the authors the following questions:

1) What is the resolution of the 39 microscopic images which the authors used in the current paper?

2) What is the pixel size of the analysed images presented in the current manuscript?

Author Response

Please see the attached file with point-by-point response. Thanks a lot.

Reviewer 2 Report

The paper presents the use of modern strategy based on image segmentation and classification to identify maceral components in photomicrographs of coal – an important issue of the coal petrography. The Authors consider the possibility of using geometric, grayscale and texture features to proposed intelligent classification. They used very useful method – Random Forest (RF) and some evaluation criteria to classification analyzed objects. This solution based on image segmentation is innovative and may be of interest to the readers of the Applied Sciences. What is also important, the Authors developed public-available software for identifying maceral, that can be very useful and effective tool for geologists. So, I recommend the paper to be published. However, before the publication I would ask the Authors to clarify the following issues:

1. How were the macerals selected for analysis? What about the other macerals, i.e. for example with funginite or macrinite in the inertinite group or macerals from the vitrinite group? I think that the article should contain information about it.

2. How do the Authors understand "region for specific maceral"? There is no clear explanation in the text.

3. How many samples were used for preliminary analyzes carried out by 5 petrographers and how the compatibility between petrographers was evaluated?

4. The problem of grayscale analysis in the point appears. It may happen that a given measurement is a noise, not a point representing a given maceral component. What in that case?

5.  How was the voting conducted? How was the final voting result determined? (Fig. 3)

6. The analyzes include maceral groups and some individual macerals. What if mineral matter is present in the samples? Were such samples examined?

7.  The analyzes are based on 172 parameters. With such a large number of parameters describing the objects, it turns out that some of them are correlated with each other. It seems reasonable to conduct appropriate analyzes and select the most important parameters from the point of assessment of the conducted analyzes.

8.Authors set the number of trees to be 300 (Fig. 7). In my opinion, the analysis of Figure 7 indicates that enough number of trees is e.g. 150 or 200, even considering the tradeoff between the computed model and the robustness of the developed model.

9. What about the quantitative content of maceral components in the tested coal samples? Such information seems to be very important for petrographers performing quantitative analysis of coal. Maybe is this a method developed only for qualitative analysis of samples?

10. One important mistake found:

· Row 81, page 2, write “Młynarczuk and Skiba” instead of “Mariusz et al.,”.

Author Response

Please check the attached file with point-by-point response. Thanks a lot.
